

# Variability in [14]C contents of soil organic matter at the plot and regional scale across climatic and geologic gradients

T. S. van der Voort[1], F. Hagedorn[2], C. McIntyre[1,3], C. Zell[1], L. Walthert[2], P. Schleppi[2], X. Feng[1,4], T. I. Eglinton[1]

[1]Institute of Geology, ETH Zürich, Sonneggstrasse 5, 8092 Zürich, Switzerland

[2]Swiss Federal Research Institute WSL, Forest soils and biogeochemistry, Zürcherstrasse 111, 8903 Birmensdorf, Switzerland

[3]Department of Physics, Laboratory of Ion Beam Physics, ETH Zurich, Otto-Stern-Weg 5, 8093 Zurich

[4]State Key Laboratory of Vegetation and Environmental Change, Institute of Botany, Chinese Academy of Sciences, Beijing 100093, China

*Correspondence to:* T.S. van der Voort (tessa.vandervoort@erdw.ethz.ch)

**Abstract.** Soil organic matter (SOM) forms the largest terrestrial pool of carbon outside of sedimentary rocks. Radiocarbon is a powerful tool for assessing soil organic matter dynamics. However, due to the nature of the measurement, extensive [14]C studies of soils systems remain relatively rare. In particular, information on the extent of spatial and temporal variability in [14]C contents of soils is limited, yet this information is crucial for establishing the range of baseline properties and for detecting potential modifications to the SOM pool. This study describes a comprehensive approach to explore heterogeneity in bulk SOM [14]C in Swiss forest soils that encompass diverse landscapes and climates. We examine spatial variability in soil organic carbon (SOC) [14]C, SOC content and C:N ratios over both regional climatic and geologic gradients, on the watershed- and plot-scale and within soil profiles. Results reveal (1) a relatively uniform radiocarbon signal across climatic and geologic gradients in Swiss forest topsoils (0-5 cm, $\Delta^{14}C=159\pm36.4$, n=12 sites), (2) similar radiocarbon trends with soil depth despite dissimilar environmental conditions, and (3) micro-topography dependent, plot-scale variability that is similar in magnitude to regional-scale variability (e.g., Gleysol, 0-5 cm, $\Delta^{14}C$ 126±35.2 , n=8 adjacent plots of 10x10m). Statistical analyses have additionally shown that $\Delta^{14}C$ signature in the topsoil is not significantly correlated to climatic parameters (precipitation, elevation, primary production) except mean annual temperature at 0-5 cm. These observations have important consequences for SOM carbon stability modelling assumptions, as well as for the understanding of controls on past and current soil carbon dynamics.

## 1 Introduction

Soil organic matter (SOM) constitutes the largest terrestrial reservoir of carbon outside of that held in sedimentary rocks (Batjes, 1996). With on-going rapid changes in land use and climate, carbon stability within this reservoir may be subject to change (Davidson and Janssens, 2006; Melillo et al., 2002; Schimel et al., 2001; Trumbore and Czimczik, 2008). The processes that lead to both stabilisation or destabilisation of SOM, however, remain poorly understood (von Lützow et al., 2008; Schmidt et al., 2011; Trumbore and Czimczik, 2008). The existence of numerous biological and physicochemical processes that operate simultaneously





confound our understanding of, and ability to accurately model SOM dynamics (Lutzow et al., 2006).

Radiocarbon measurements of SOM in combination with compositional data are increasingly used to assess SOM turnover rates and residence time, often coupled with modelling (Davidson and Janssens, 2006; Schrumpf et al., 2013; Sierra et al., 2012; Tipping et al., 2010, 2011; Torn et al., 2005; Trumbore, 2009). The marked $^{14}$C enrichment associated with the radiocarbon "bomb-spike" (Stuiver and Polach, 1977) aids in in the elucidation of processes taking place on decadal scales, while natural $^{14}$C decay can shed light on processes that occur on timescales of centuries to thousands of years. Although a powerful tracer, it is presently not well established how the amplitude of $^{14}$C variations associated with carbon turnover and storage are masked by spatial heterogeneity. Soils are intrinsically highly heterogeneous over a range of spatial scales (Goovaerts 1998), and effects of climatic, geological and topographical gradients as well as small-scale variability at the plot or pedon scale are often overlooked or generalized. While spatial variability in soils has been explored in the context of SOM compositional parameters (Conen et al., 2004; Goovaerts, 1998; Grüneberg et al., 2010), systematic studies of its influence on $^{14}$C characteristics of SOM are lacking. Moreover, despite the high carbon stocks in subsoils, deeper soil carbon characteristics have been much less intensively compared to topsoils, and therefore remain poorly understood, particularly in the context of $^{14}$C variability (Fontaine et al., 2007; Rumpel and Kögel-Knabner, 2011).

In order to examine the temporal and spatial scales of $^{14}$C variability in soils, we have investigated SOM $^{14}$C signatures in depth profiles across diverse climatic, and geologic gradients, as well as over different spatial scales. These findings provide a framework for anticipating and accounting for spatial variability that are of importance in developing sampling, measurement and carbon-cycle modelling strategies. The broader goal is to provide the foundation for understanding, and ultimately predicting, regional-scale SOM (in)stability based on radiocarbon characteristics.

## 2   Materials and Methods

### 2.1   Study area and plot selection

The study area, which covers the entirety of Switzerland, is located between 46-47°N and 06-10°E with an accompanying elevation gradient between roughly 480 and 1900 m above sea level (a.s.l.). Within this geographic and altitudinal range, Mean Air Temperature (MAT) and Mean Annual Precipitation (MAP) varies from 1.3 to 9.8 °C and ~ 600 to 2100 mm y$^{-1}$, respectively (Etzold et al., 2014; Walthert et al., 2003). Samples investigated in this study derive from the Long-term Forest Ecosystem Monitoring (LWF) network monitored and maintained by the Swiss Federal Research Institute of Forest, Snow and Landscape Research (WSL) (http://www.wsl.ch/lwf). The LWF sites are all located in mature forests and samples were collected in the during the 1990s (Innes, 1995). An additional sampling campaign was conducted at the same sites in 2014. They encompass marked climatic and geologic gradients, yielding an array of different soil types (Fig. 1). The LWF-network was chosen because it features i) archived samples ii) environmental and ecological monitoring data and iii) long-term monitoring and sampling potential. An overview of the most relevant lithological, geomorphic and



climate variables of the sampled sites can be found in Table 1. Soils are classified in accordance with the World
Reference Base (WRB, 2006).

### 2.2    Sampling methodology

Samples were taken in the course of the 1990s in a 43 by 43 meter (~1600 m$^2$) plot, that was divided in
quadrants of 20 meters (~400 m$^2$) and sub-quadrants of ten meters (~100 m$^2$), each with 1 m of walking space in
between (Walthert et al., 2002; Fig. 2A). For depth increments down to 40 cm soil of an area 0.5 by 0.5 m was
sampled (0.25 m$^2$), for samples > 40 cm corers were used (~2×10$^{-3}$ m$^2$). Because different soil types were
included, depth increments rather than genetic horizons were analysed. In order to consider the plot-scale spatial
variability, three mixing schemes were devised.

1.  Eight discrete samples per depth were taken from the sub-quadrants according to a chessboard-pattern
80       (Fig. 2A)

2.  Sixteen samples per depth were taken from sub-quadrants, and combined and mixed for each quadrant
    of the plot (1-4, 5-8 etc), yielding in total four composite samples (Fig. 2B).

3.  Sixteen cores are taken from sub-quadrants, all samples are combined and mixed to yield a single
    composite sample (Fig. 2A)

The destructive nature of digging soil profiles down to the unweathered parent material only allowed for single
profiles to be taken proximally to the intensively monitored plot.

Variability within and between two similarly sized plots was investigated in a subalpine spruce forest in the
Alptal Valley (central Switzerland), separated by approximately 500 m. The plots are located within the same
watershed and hosted on the same bedrock and are primarily distinguished by their degree of tree coverage.

The two plots concerned are: (1) WSL LWF plot (~1600 m$^2$, sampled 1998), covered mainly by Norway Spruce
trees. Sampling was done by depth-increment in a 10×10 m grid (Fig. 2B); (2) WSL NITREX control plot,
(~1500 m$^2$, sampled 2002) with a relatively open tree canopy with well-developed herbaceous vegetation and
shrubs (Schleppi et al., 1998). This site was sampled by horizon (litter, topsoil and Gleyic horizon) on an 8×8 m
grid, yielding 24 soil cores. These cores were grouped according to their morphology.

Interpolation based on documented soil cores was used to convert the NITREX control plot horizon-logged cores
to depth-logged cores.

Soil samples were dried in an oven at 35-40°C, sieved through 2 mm, and stored in a climate controlled storage
facility at the WSL (Walthert et al., 2002).

### 2.3    Elemental, $^{13}$C and $^{14}$C analysis

Vapour acidification (12 M HCl) of soils in fumigated silver capsules was applied to remove inorganic carbon in
the samples (Komada et al., 2008; Walthert et al., 2010). Carbonate-poor samples were acidified for 24 hours at
room temperature, for carbonate-rich samples this treatment was extended to 72 hours at a temperature of 60 °C.
Both acidification treatments were tested on the same sample and no significant effect on the radiocarbon values





was found. To ensure the removal of all carbonaceous material, $\delta^{13}C$ values of the soil samples were measured.

All glassware and silver capsules were combusted at 450 °C (6 h) prior to use.

Graphitization of the soil organic matter was performed using an EA-AGE (elemental analyser-automated graphitization equipment) system. (Wacker et al., 2009). The radiocarbon content of the graphite targets was measured on a MICADAS (MIniturised radioCArbon DAting System) at the Laboratory of Ion Beam Physics, ETH Zürich (Wacker et al. 2010). Samples were calibrated against oxalic acid II (NIST SRM 4990C) and an in-

house anthracite coal as well as an in-house soil standard (Aptal soil 0-5 cm). Intra-sample (Alptal soil 0-5 cm) repeatability is high with a value and standard deviation of Fraction Modern (Fm) = 1.1263 ±0.0003696 (n=12).

An Elemental Analyser-Isotope Ratio Mass Spectrometer system (EA-IRMS, Elementar, vario MICRO cube – Isoprime, Vison) was used to measure the absolute concentrations of carbon and nitrogen, as well as stable carbon isotopic composition ($\delta^{13}C$). Calibration standards used for the C and N concentrations, and C:N ratio

were peptone (Sigma) and atropine (Säntis) and only atropine (Säntis). The $\delta^{13}C$ values were used to assert that all inorganic carbon has been removed by the procedure, as well as provide a confirmation of the signature of the plant input.

### 2.4 Turnover model and statistical analysis

For the turnover time or mean residence time (MRT) estimates the system was assumed to be in steady state,

with no time lag between atmospheric carbon fixation by plants and conversion to soil carbon. A time-dependent model (Eq. (1)) (Herold et al., 2014; Torn et al., 2009) was used in combination with atmospheric data of Levin et al. (2010). Here, $k$ refers to the decomposition rate constant and $\lambda$ to the radioactive decay constant of $^{14}C$ in $y^{-1}$ (1/8267). $R_{sample}$ is inferred from $D^{14}C$ as shown in Eq. (2) (Herold et al., 2014; Solly et al., 2013).

$$R_{sample} = k * R_{atm,t} + (1 - k - \lambda) * R_{sample(t-1)} \qquad (1)$$

$$R_{sample} = \Delta^{14}C \frac{\Delta^{14}C_{sample}}{1000} + 1 \qquad (2)$$

MRT was calculated for the samples for which the signal was above the atmospheric $D^{14}C$ signal of the sampling

year, indicating a predominant input of post-bomb derived carbon. The constant $k$ is found by matching the modelled and measured values of Eq. (1), and MRT is obtained by inverting $k$ (Torn et al., 2009).

Statistical analyses were performed in R, version 3.1.2. Spatial variability of was quantified by calculating the coefficient of variation (CV) of the Fm $^{14}C$ values. The Fm values rather than the $D^{14}C$ values were used as CV tends toward infinity if the average value of the variable concerned nears to zero, which in this case would cause

unwanted bias (Stuiver et al., 1997). Drivers of CV were analysed by Pearson correlation (rcorr function; hmisc package) using ancillary climate and geographic data taken from the WSL LWF network (Walthert et al., 2003). Ancillary data entailed MAT, MAP, net primary production (NPP), pH and mineralogical composition such as clay, sand and silt content. The Spearman correlation coefficient (rcorr function; hmisc package) was used to test





relationships between Fm and climatic and compositional parameters for data from separate depth intervals. The

effects of climate and soil properties on radiocarbon ($D^{14}C$) was also tested by fitting mixed-effect models by maximum likelihood (lme function; nlme package; (Pinheiro and Bates (2000)). Mixed-effect model analyses provide an additional benefit because they eliminate variance introduced by random variables, whereas this is lacking in e.g. the Spearman correlation. For the mixed-effect model, the complete dataset was used, i.e. all depths, soil cores and all individual depths were included. The variables *site* and *core* were always taken to be

random effects following the sampling design at the LWF sites. Environmental drivers (MAT, MAP, NPP, pH, clay content) were taken as fixed effects and tested individually. For the analysis concerning all depths, $D^{14}C$ was tested against depth and the additional parameters. For set depth intervals, $D^{14}C$ was related only to one parameter per analysis. In all final models, normality and homoscedasticity of the residuals were verified visually with diagnostic plots; the dependent variables were all log transformed in some cases to meet

assumptions of normality and variance homogeneity. Semivariograms (variog function; geoR package) were also used to assess the patterns of variability with distance.

## 3 Results

Spatial variability was systematically investigated from the regional (100-300 km) to the plot (10-100 m) scale.

### 3.1 Regional scale (100-300 km) variability

At the LWF sites across Switzerland (Table 1), the standard deviation in $\Delta^{14}C$ values of the organic (LF) (n=12), 0-5 cm (n= 12) and 5-10 cm (n=11) layers exceeds the instrumental error by up to tenfold with values of 36.4, 28.6 and 34.9 ‰, respectively (Table 2). Furthermore, the coefficient of variation in SOC content increases with depth whilst $\Delta^{14}C$ variability remains almost constant (Fig. 3). The MRT relative error remains constant with depth (Table 2). For the former and latter analyses, a Podzol site (Beatenberg) was excluded because its unique

thick organic layer distorted the topsoil age. Spearman correlation identified few significant correlations between $\Delta^{14}C$ and climatic variables. Exceptions include a negative correlation between $\Delta^{14}C$ versus MAP and versus SOC in the 10-20 cm depth interval samples and a positive correlation between $\Delta^{14}C$ and relief (slope) in samples from 0-5 cm depth (Table 5). Mixed-effect models reveal few significant effects of environmental variables on $\Delta^{14}C$ at the 12 LWF sites except SOC (but not at 5-10 cm) and MAT at the 0-5 cm depth interval

(Table 6).

### 3.2 Plot-scale (10-100 m) variability

In the subalpine spruce forest (Gleysol, Alptal catchment) two similarly sized plots were compared to assess inter-plot and intra-forest variability. Results show essentially similar ranges and trends with soil depth for both sites, both in terms of $^{14}C$ and C:N ratio. In the deeper soil horizons, the NITREX site has generally lower but

also more variable $\Delta^{14}C$ values and slightly lower C:N ratios. Due to the differences in horizon versus depth sampling procedures further statistical analysis is however not possible (Fig. 4).

Plot-scale spatial variability was investigated at three LWF-sites exhibiting the greatest (Podzol and Gleysol) and least (Cambisol) heterogeneity in terms of micro-relief (Fig. 5). Based on these end-members, we set brackets on





the minimum and maximum spatial variability in composition that we expect over the entire suite of study sites.

These measurements reveal that on a scale of ten meters, standard deviation associated with intra-site variability in $\Delta^{14}C$ values can exceed analytical error by tenfold. In the selected plots, the greatest within-plot variability in $\Delta^{14}C$ values among the eight 0.5 by 0.5 m soil pits, each taken in subquadrants of 10 x 10 m (Fig. 2), ranges from 50 ‰ (worst case scenario, Podzol) to 20 ‰ (best case scenario, Cambisol) (Table 3). Slope is positively correlated with the fraction modern coefficient of variation ($CV_{Fm}$) of single cores for the three high-resolution

(Cambisol, Gleysol, Podzol) sites ($r^2 = 0.99$ for n =3). The SOC concentration in the single cores of the Cambisol and Podzol correlate positively with $\Delta^{14}C$ signature (resp. $r^2 = 0.78$, n =7 and $r^2 = 0.60$, n =8), but for the Gleysol there is no correlation ($r^2 = 0.00$) (Fig. 5). Pearson correlation between variability (expressed as $CV_{Fm}$) and both environmental variables (MAT, NPP, Elevation, Slope, SOC content, C:N ratio) yielded weak correlations ($r$ <0.35), except for clay content ($r = 0.71$) and MAP ($r = 0.51$).

In order to assess the efficacy of mixing of samples to achieve representative samples, four composite samples from the quadrants were measured (each ~400 m$^2$) as well as a composite of 16 samples (encompassing the entire plot, ~1600 m$^2$) as shown in Fig. 5. For the Cambisol and Podzol, the difference in $\Delta^{14}C$ values between physically mixed bulk samples and the statistical average of individual samples is close to or within analytical uncertainty (resp. 15.1 and 0.46 ‰, Fig. 5) and considerably smaller than the standard deviation of the quadrant

samples. Semi-variograms did not yield satisfactory results, as the variance did not plateau on the involved spatial scales.

### 3.3 Influence of micro-topography and waterlogging on $^{14}C$ variability

The assessment of within-plot variability related to micro-topography was investigated at the NITREX plot. Cores were grouped according to morphological characteristics of the terrain:

(1) Mounds (relative elevation up to 50 cm) with deeper water table, with a mor type organic layer and an oxic Gley horizon.
(2) Weak depressions (relative depression up to 15 cm) with a shallow water table, an anmoor-type organic layer and a partly (mottled) anoxic Gley soil.
(3) Stronger depressions (relative depression up to 25 cm) with a very shallow water table, an anmoor-type

organic layer and underlying soil that is characterized by an anoxic Gley horizon.
(4) Intermediate type (between type (1) and (2)) with flat top, intermediate water table and anoxic Gley horizon.

Radiocarbon data combined with SOC contents and C:N ratios show (Fig. 6) that (1) contrasts exist between soils developed under different microtopographic features with the mound (oxic Gley) having the highest $\Delta^{14}C$

value in the gley layers, indicating a higher degree of incorporation of organic matter containing bomb-derived radiocarbon. The strong (anoxic Gley) and intermediate depression (partially anoxic Gley) exhibit very similar radiocarbon signal in the organic layer, but with depth show increasingly lower $\Delta^{14}C$ values on the mound, with the weak depression being the most $^{14}C$ depleted in the Gley. $\Delta^{14}C$ values for the intermediate soil type falls between mounds and depressions. Additionally (2) SOC concentrations are higher in the topsoil of the mounds,

but a reversed trend occurs at greater depth where the weak depression has higher SOC contents in the Gley





horizon. Lastly (3) C:N ratios show a similar reversal, as the weak depression has a markedly lower C:N value than the other micro-topographic features in the topsoil, while having a higher value in the gleyic horizon.

### 3.4 Spatial variability with depth

In order to establish $^{14}$C variability as a function of depth in topsoils, four composite soil samples (designed to
assess within-plot spatial variability, each composed of four sub-cores from ~400 m$^2$, that encompass a total plot area of ~1600 m$^2$, Fig. 2) were examined for five sites, each comprising different soil types, at three depth intervals (0-5, 5-10 and 10-20 cm) (Fig. 7). These analyses revealed that spatial variability, expressed in SOC content and $\Delta^{14}$C, is more muted but persists at depth. For the four 400 m$^2$ composites, the standard error of spatial variability is only 1 to 3 times the analytical error, with the exception of the highest variability end-
member (Podzol). The use of composite samples helps in reducing the potential for anomalous and unrepresentative values (Fig. 5, 7). For examination of deeper soils (> 20 cm), samples taken along a single profile per site were measured (Fig. 8). Results show that regardless of marked differences in soil type (and associated processes that influence SOM) and large climatic gradients, the depth gradient in $^{14}$C and overall amplitude of $\Delta^{14}$C variations are remarkably similar. Two main exceptions are the Podzol, whose $\Delta^{14}$C value in
the topsoil is markedly lower than the other soils, and the erratic behaviour of the $\Delta^{14}$C trend in the Cambisol below 1 m depth. When taking into account the errors associated for single samples in the topsoil (Cambisol, Podzol, Gleysol), we see that the range of $\Delta^{14}$C values overlaps for four soils (Cambisol, Podzol, Calcisol and Luvisol) within one standard error (Fig. 8). Regressing the $\Delta^{14}$C values with soil depth to 100 cm depth showed tight linear relationships (r=0.93-0.99). The depth gradients ranged between -3 and -8.5‰ cm$^{-1}$ for the five sites,
but showed no consistent relationship with any of the environmental variables (Table 1). The standard deviation associated with the SOC content and C:N ratios at these three locations also exceeds the values for these parameters in the five profiles. Additionally, with the exception of high-variability end-member Podzol, variability of $^{14}$C values with depth exceeds lateral variability for the intervals 0-20 cm. $^{14}$C variability with depth for deeper soils (>20 cm) exceeds lateral variability for all soil types (Table 4).

## 4 Discussion

### 4.1 Uniformity in $^{14}$C signature across geological, climatic and depth gradients

Understanding the inherent stability and resilience of SOM in different climates is key for predicting how the carbon cycle may respond to future changes. The uniformity in topsoil radiocarbon signatures across the 12 sites spanning marked climatic and geological gradients as well as soil types implies that the MRT of SOM in
different ecosystems is relatively similar despite dissimilar environmental conditions. Few significant relationships emerged from comparison of $\Delta^{14}$C values with environmental characteristics of the LWF sites (Tables 5 and 6). The significant positive correlation between MAT and $\Delta^{14}$C for 0-5 cm depth found in the mixed-effect model (Table 6) indicates that the speed of incorporation of bomb carbon is positively related to MAT, which is consistent with the findings of Leifeld et al. (2015) in grassland soils. However, the lack of
correlation in the organic layer and deeper layers indicate that confounding factors such as variable ages from above- and belowground litter input (Solly et al., 2013; Schrumpf et al., 2013) may impact the $\Delta^{14}$C signal. The





strong negative Spearman correlation of $\Delta^{14}C$ and MAP at 10-20 cm depth implies a slower turnover which may caused by increased waterlogging or anoxic conditions induced by higher precipitation. The significant negative correlation of $\Delta^{14}C$ and SOC content at 10-20 cm depth may reflect dilution of bomb-$^{14}C$ derived carbon or delayed transmission of carbon containing the bomb-$^{14}C$ signal in SOC-rich soils. The positive relation between $\Delta^{14}C$ and relief (slope) for the shallowest (0-5 cm depth) samples may be due to erosion-induced higher turnover rates that result in higher incorporation of bomb-derived material. The lack of significant correlation between $\Delta^{14}C$ and precipitation and primary production indicates that the incorporation of carbon in soils is not consistently dependent on these climatic factors. This is consistent with findings of Doettler et al. (2015) that found that factors other than climate exert strong controls on carbon stocks. It has to be notes that the former study concerned stocks only (i.e. not turnover), but when assuming a steady state system, it is reasonable to assume that the speed of incorporation of carbon and hence turnover is directly related to carbon stocks. Furthermore, the high overall similarity in radiocarbon profiles with depth for five strongly contrasting soil types implies that this similarity persists with depth. The limited size of the data set prohibits more rigorous statistical analyses. This uniformity also indicates an insensitivity of $^{14}C$ in SOM to environmental conditions. One explanation for this apparent insensitivity could be that C input into soils and decomposition are driven by similar factors; for instance increased litter input with increasing MAT is accompanied by an increased decomposition rate.

The $\Delta^{14}C$ value for the Podzol in the topsoil is lower than for all other sites, which may be due to the unusually thick (>20 cm) organic layer, which delays the transfer of the $^{14}C$ signal to the mineral soil. The erratic behaviour below 1 m depth in the Cambisol (Lausanne) may be derived from a mixture of syn-glacial and post-glacial carbon present in the moraine underlying the soil. The relatively small variability at the regional scale was even more striking as the corresponding CV was almost the same as at the plot scale for soil samples located approximately 10 m apart from each other (Table 3).

There are only a few comparisons of $^{14}C$ characteristics among soil types, but other studies have reported similar trends to those shown in Fig. 8 (Baisden and Parfitt, 2007; Jenkinson et al., 2008). Scharpenseel and Becker-Heidelmann (1989) however, find very different $^{14}C$ ages between soil types for the depths 20, 50 and 100 cm on different continents. Mills et al. (2013) analysed literature data for forest top soils and find a similar range in values to our samples. When comparing to cultivates soils we found large differences in $\Delta^{14}C$ signature at depth (Paul et al., 1997). The variable time-point and horizon-notation of other soil radiocarbon samplings limit further comparison to existing literature (Eusterhues et al., 2007; Paul et al., 1997; Rumpel and Kögel-Knabner, 2011; Scharpenseel et al., 1989). The time of soil formation may be paramount to the differences that develop in the bulk $^{14}C$ age of organic carbon over time. Soil formation for the soils studied here started after the last glacial retreat and can hence be assumed to have started to form around the same time, which may explain the similarity in their $^{14}C$ distribution with depth. Overall, the high similarity of $\Delta^{14}C$ signatures between disparate soil types even at depth indicates that the relative independence on climatic parameters may persist in deeper soils, and hence also for older SOM and during long-term C storage.





Further insights into the stability and vulnerability of SOM will likely require investigations of specific fractions (Schrumpf et al., 2013) and compounds (Feng et al., 2008), where contrasts in the dynamics of SOM will likely

be amplified. Soil carbon turnover and stability models (e.g. CENTURY, Yasso, SoilR) that are highly dependent on input parameters would also greatly benefit from improved estimates of spatial variability in $^{14}$C and other properties at the plot and regional scale, in combination with site-specific biochemical data (Liski et al., 2005; Schimel et al., 2001; Sierra et al., 2014).

### 4.2    Plot-scale variability and the influence of micro-topography

The marked plot-scale variability in $^{14}$C revealed in this study is similar in magnitude to $^{14}$C signal variations observed across larger-scale gradients. In light of this observation, developing and implementing sampling strategies to account for small-scale heterogeneity in SOM is essential for robust recognition and assessment of changes in SOM composition and stocks. It is also crucial that we better understand factors that drive small-scale variability, as only then will we be able to incorporate these factors in larger-scale models of SOM turnover. We

find a positive correlation between $\Delta^{14}$C and SOC content when examining the 0-5 cm intervals for individual samples from the 10 x 10m sub-quadrants (Fig. 5). When looking at the variability between composite cores at 0-5 and 5-10 cm, this relationship disappears and re-appears only at 10-20 cm depth (Fig. 8). However, the relationship between $\Delta^{14}$C and SOC differed among sites, indicating that other factors than C pool size contributed to the small scale variability (Fig. 5). One reason could be spatially varying $^{14}$C inputs from plants

which as a consequence of differences plant functional types, plant/tree age, and between plant components such as roots, stems, needles and leaves (e.g. Solly et al., 2013; Fröberg et al., 2012). The Cambisol, which exhibits a tight relation between $\Delta^{14}$C and SOC, is located in a relative homogeneous beech forest. In contrast, there is a poor correlation between $\Delta^{14}$C and SOC in the Gleysol, and is characterized by a more complex vegetation mosaic with annual herbs in depressions and perennial dwarf shrubs as well as spruce trees on the mounds

(Thimonier et al., 2011). $\Delta^{14}$C values of annual plants may differ substantially from perennial woody plants (Solly et al., 2013), and this could impart significant difference in $^{14}$C input signals among microsites. The Podzol site has an open canopy structure with dwarf shrubs dominating in the understory which may deliver an different $^{14}$C input signal than 'aged' needle litter from trees. For the three high-resolution sites (Podzol, Gleysol, Cambisol) we observed that $^{14}$C variability was particularly high at sites with steeper slopes, which are also

marked by larger differences in micro-topography as well as higher occurrence of soil disturbances induced by felled trees (windthrow). We thus attribute the high variability in $\Delta^{14}$C values, especially for the Podzol and Gleysol, to soil-mixing induced by tree-throw and subsequent higher erosion of loosened soil particles. Micro-topographic contrasts can also be amplified by soil erosion pathways (rills, ditches) affecting the hydrological properties such as the degree of waterlogging and water routing.

Geostatistical methods such as semi-variograms were applied to the data but yielded unsatisfactory results because the variograms did not reach constant values. In terms of micro-topography, based on field observations, the measured plots are representative for the larger areas. However, sampling and analysing a larger plot-size would be informative for a semi-variogram analysis.





In the subalpine spruce forest (Gleysol, Alptal catchment), clear differences in radiocarbon signature can be seen between different micro-topographic features (Fig. 6), confirming prior observations of compositional variations (Schleppi et al., 1998). As would be expected, mounds that are conducive to less waterlogged (more oxygenated) conditions contains more bomb $^{14}$C-containing (younger) carbon as root and above-ground inputs are higher in these areas. The anoxic depressions would expected to receive the least input of biomass and preserve the material most optimally compared to more oxygenated soil types. However, contrary to this expectation, the 'intermediate' weak depression with a mottled oxygenated gley horizon, consistently holds the lowest $\Delta^{14}$C signature and exhibits the highest C:N ratios within the gley horizon despite similar SOC contents to the other microsites. A lower C:N ratio is widely regarded as a proxy for a higher degree of SOM degradation (Conen et al., 2008). Overall, the geochemical characteristics are indicative of decreased carbon degradation in the partially oxic site compared to the more fully oxic or completely anoxic sites. This may reflect different responses of microbial communities to oscillating redox conditions or other properties associated with intermittent waterlogging. In this context, differences in hydrologic flow path (vertical, lateral) and residence times (flow versus stagnation) may be important. An additional factor induced by micro-topography is the vegetation-induced differences in lag-time of the transfer of the atmospheric radiocarbon signal to SOM. The depressions support growth of vascular plants with annual shoots and younger fine roots, which may transfer the atmospheric signal faster than mound areas dominated by spruce trees and shrubs (Solly et al., 2013).

## 5 Conclusions

Variability of SOM signatures, notably radiocarbon, in shallow and deep horizons of soils spanning large geologic and environmental gradients is similar to that observed at the plot-scale. Spatial variability could not be linked to clear environmental drivers (climatic and soil properties), except clay content and MAP. On a regional scale, there was also no apparent strong driver for $\Delta^{14}$C, except a weak relationship with MAT for the shallowest (0-5 cm depth interval) soil horizons. While the present observations remain limited in geographic scope, the relative homogeneity of $\Delta^{14}$C signatures observed in surface and deep soils across climatic and geologic gradients implies that the speed of C incorporation may be relatively insensitive to changing climate conditions. The strong similarities in radiocarbon trends with depth among the different sites support the validity of the extrapolation of currently sparse $^{14}$C observations from the topsoil to deeper soil horizons in carbon modelling studies. Systematic sampling of soils over areas that incorporate the micro-topographic variability will reduce the impact of outliers and yield representative samples. The latter is essential for the use of radiocarbon to assess carbon turnover and associated processes in forest soils. Future $^{14}$C investigations should include expansion of assessments of spatial variability to a broader range of soils, extension of such comparisons to deep soil horizons, and in-depth investigations of variability among SOM fractions and individual compounds.

## 6 Author Contribution

T.S. van der Voort selected the samples, performed the measurements and large portion of the statistical analysis. F.Hagedorn coordinated the statistical analysis and provided scientific feedback. C. McIntyre coordinated the $^{14}$C measurements and associated data processing. C. Zell aided in the measurements of samples for $^{14}$C. L. Walthert and P. Schleppi collected samples or coordinated sample collection. X. Feng contributed to



the project set-up and planning. T. Eglinton provided the conceptual framework and aided in the paper structure set-up. T.S. van der Voort prepared the manuscript with help of all co-authors.

## 7 Acknowledgements

We thank the Swiss National Science Foundation (SNF) and the National Research Plan 68 „Sustainable use of
soils as a resource" (NRP68) for funding this project (SNF 406840_143023/11.1.13-31.12.15) in the framework
of the SwissSOM „Sentinels". We thank the WSL LWF team for the sampling and data acquisition, and we
would also like to thank the Laboratory of Ion Beam Physics group of ETH Zurich, specifically Lukas Wacker
for enabling the measurements. Many thanks also to the other SwissSOM team members in this project: Sia
Gosheva, Beatriz Gonzalez and Cedric Bader for their constructive input and synergy.  Many thanks also to
Emily Solly for her helpful comments and input.

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

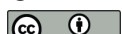



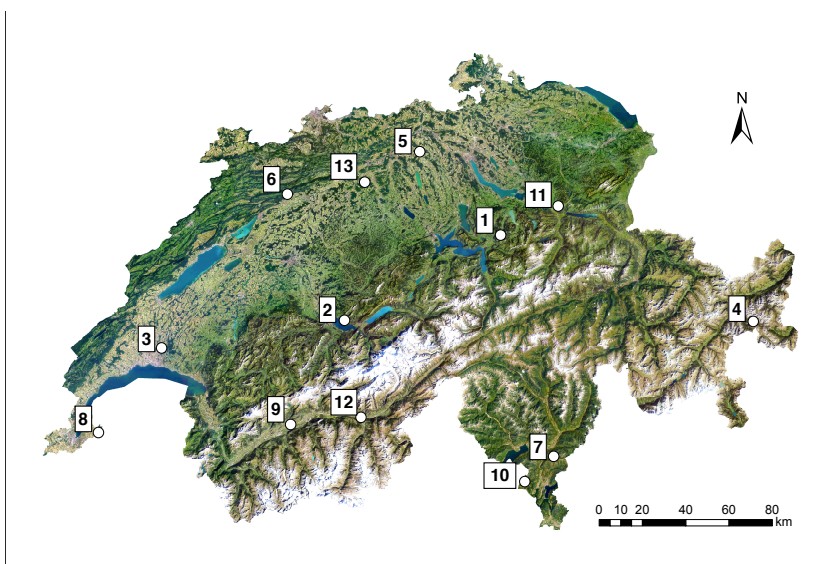

**Fig. 1.** Locations of the sampled sites within Switzerland indicated by white points. Numbers refer to the locations [1]Alpthal, [2]Beatenberg, [3]Lausanne, [4]Nationalpark, [5]Othmarsingen, [6]Bettlachstock, [7]Isone, [8]Jussy, [9]Lens, [10]Novaggio, [11]Schaenis, [12]Visp and [13]Vordemwald (Innes, 1995).




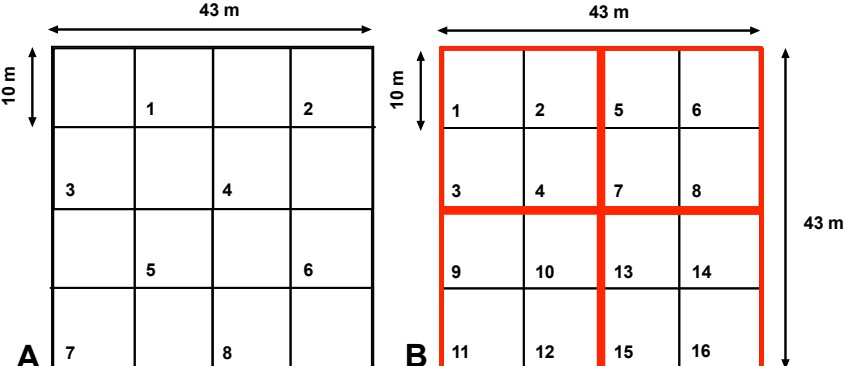

**Fig. 2**. Soil sampling scheme according to Walthert et al., 2002 at the Long-Term Forest Ecosystem (LWF) Research Program, (A) 8 individual samples taken in a chessboard pattern, (B) 16 samples that are combined to mixed samples in each plot quadrant, as indicated by the red boxes (1-4, 5-8 etc).






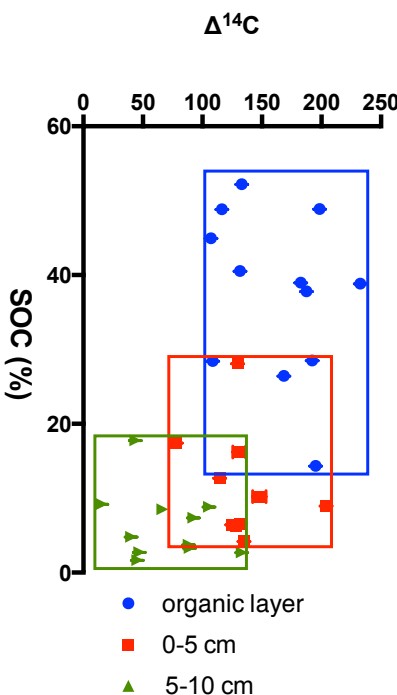

**Fig. 3.** Relationship of $\Delta^{14}$C versus soil organic carbon content (SOC) on the regional scale within Switzerland (n=12 sites), with each sample representing the average of four cores covering an area of 400 m$^2$. Samples were taken in between 1995-1998. Trends show decreasing variability of soil organic carbon content with depth, but no significant changes in the variability of $\Delta^{14}$C contents with soil depth.





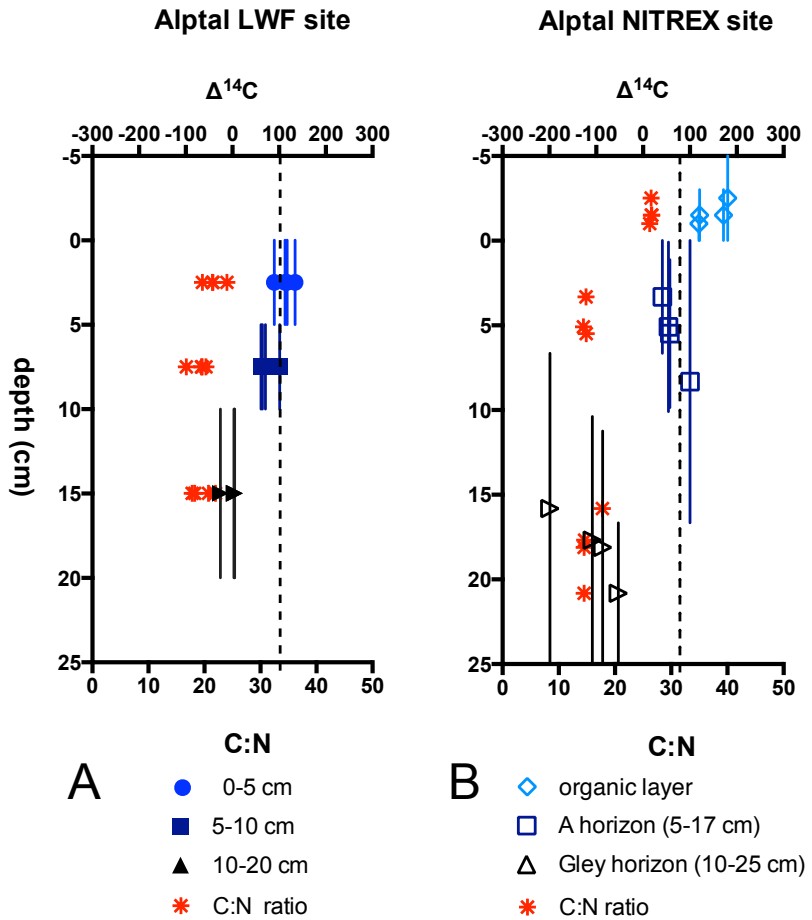

**Fig. 4.** Intra-forest variability (A) four samples each taken from an area of approximate 400 m² totaling to 1600 m² (B) four samples representing micro-topographic end members sorted by horizon covering around to 1500 m². The vertical bars represent the depth intervals. The dashed line indicates the $\Delta^{14}C$ signature of atmosphere at the time of sampling in resp. 1998 (A) and 2002 (B) (Hua et al., 2013).






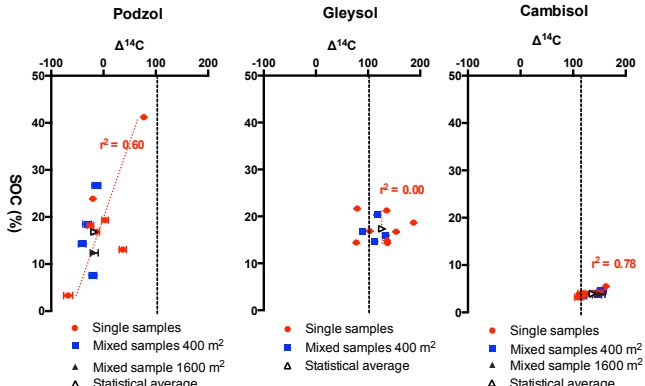

**Fig. 5.** Relationship of $\Delta^{14}C$ versus soil organic carbon (SOC) content in topsoil samples (0-5 cm) from three sites with decreasing micro-topographic relief from left to right. Error bars indicate analytical error. The pattern shows that single samples can be unrepresentative but careful physical homogenization of soil samples yields almost identical values than the arithmetic average of individual samples. The dashed vertical line in black indicates the $\Delta^{14}C$ signature of the atmosphere at the time of sampling (Hua et al., 2013). The dotted lines represent the linear regression of $\Delta^{14}C$ and SOC contents for single sample measurements.





**Fig. 6.** Variability in SOC, $\Delta^{14}$C and C:N values between separate micro-topography end-members (0-25 cm depth, in three intervals) of soils from the Alptal Valley. Weak depressions yield the lowest $\Delta^{14}$C values, whilst the mounds are most enriched in $^{14}$C. Analytical errors are smaller than symbol sizes and are not visible (between 6-8 ‰ $\Delta^{14}$C). The dotted vertical line for $\Delta^{14}$C indicates the atmospheric input at the time of sampling (2002) (Hua et al., 2013).



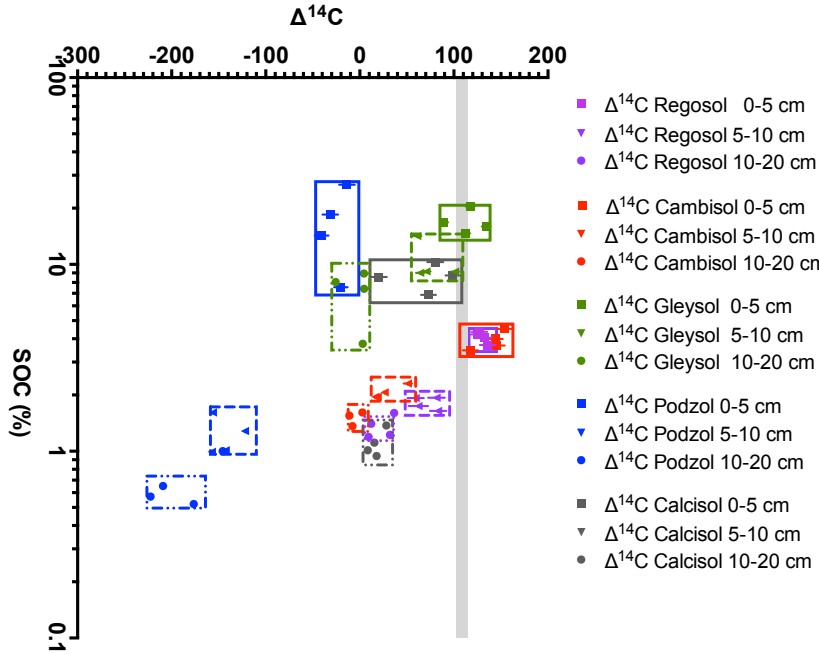


**Fig. 7.** Spatial variability with depth (composites of each 4 cores, covering 400 m$^2$) for five different soil types (Regosol, Cambisol, Gleysol Podzol and Calcisol) covering a gradient in elevation and average air temperature (MAT 9.2-1.3 °C, elevation 800-1900 m). The $\Delta^{14}C$ signature of atmospheric input at the time of sampling in indicated by grey shading (Hua et al., 2013). Variability in radiocarbon contents persists with depth and overlaps

among soil types, while carbon contents differ among soil types and differences decrease markedly with depth.





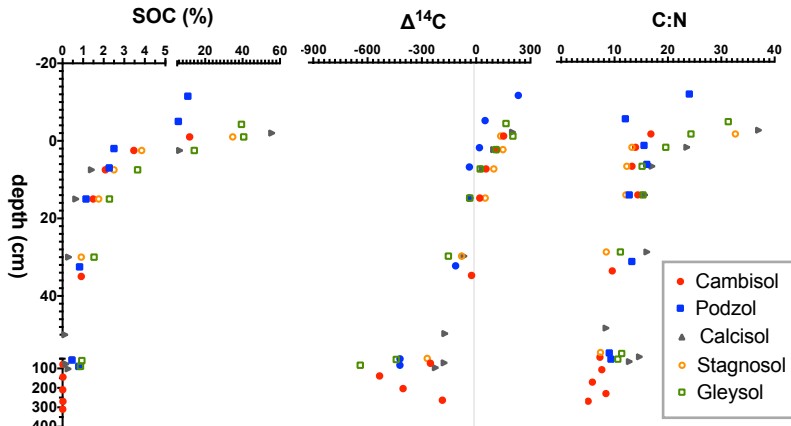

**Fig. 8.** Soil organic carbon (SOC) content, $\Delta^{14}$C and C:N ratio's for five depth profiles of distinct soil types (Cambisol, Podzol, Calcisol, Stagnosol and Gleysol) covering a climatic and altitudinal gradient. Radiocarbon trends are highly similar for four out of five soil types, while carbon content and C:N ratio varies for each soil type. The Podzol shows a different trend, with $\Delta^{14}$C consistently lower in the topsoil. Analytical errors are smaller than symbol sizes and are not visible (between 6-8 ‰ $\Delta^{14}$C).



**Table 1**. Overview of WSL LWF sampling locations and main climate variables. Mean Average Air Temperature (MAT) and Mean Average Precipitation (MAP) and Net Primary Production (NPP) measured between 1980 and 2010. Numbers in first column correspond to site numeration of Fig. 1. Soil classification in accordance with WRB (2006), altitude in meters a.s.l., soil and slope as a percentage of 90° (Etzold et al., 2014; Walthert et al., 2003).


| | Soil type[1] | Geology[1] | Location[1] | Slope[1] | Altitude[1] | MAT[2]°C | MAP[2] mm y[-1] | NPP[2] g C m[-2] |
|---|---|---|---|---|---|---|---|---|
| 1 | Gleysol | Flysch | Alpthal | 11.5 | 1149-1170 | 5.3 | 2126 | 445 |
| 2 | Podzol | Sandstone | Beatenberg | 16.5 | 1490-1532 | 4.7 | 1163 | 302 |
| 3 | Cambisol | Moraine with calcareous and shale fragments | Lausanne | 3.5 | 800-814 | 7.6 | 1134 | 824 |
| 4 | Calcisol | Alluvial fan of calcareous character | Nationalpark | 5.5 | 1890-1907 | 1.3 | 864 | 111 |
| 5 | Regosol | Moraine of calcareous character | Othmarsingen | 13.5 | 467-500 | 9.2 | 1024 | 845 |
| 6 | Cambisol | Calcareous sediments | Bettlachstock | 33 | 1101-1196 | 6.4 | 1273 | 530 |
| 7 | Cambisol | Moraine | Isone | 29 | 1181-1259 | 7.2 | 2126 | 490 |
| 8 | Stagnosol | Moraine of calcareous character | Jussy | 1.5 | 496-506 | 9.4 | 961 | 504 |
| 9 | Calcisol | Colluvial deposits | Lens | 37.5 | 1033-1093 | 8.1 | 969 | 289 |
| 10 | Umbrisol | Moraine | Novaggio | 34 | 902-997 | 9.5 | 1545 | 580 |
| 11 | Cambisol | Calc supported conglomerate | Schaenis | 30 | 693-773 | 8.5 | 1691 | 770 |
| 12 | Regosol | Calc supported colluvial deposit | Visp | 40 | 657-733 | 9.8 | 595 | 86 |
| 13 | Stagnosol | Moraine | Vordemwald | 7 | 473-487 | 8.1 | 1123 | 774 |





**Table 2.** Overview variability of $\Delta^{14}$C and standard deviation across regions and horizons converted to mean residence time (MRT) and coefficient of variation (CV).

|  | LF (n = 12) | O-5 cm (n = 12) | 5-10 cm (n = 10) |
|---|---|---|---|
| $\Delta^{14}$C | 159± 36.4 | 130±28.6 | 67.4±35.9 |
| Fm | 1.17±0.037 | 1.14±0.029 | 1.08±0.035 |
| $CV_{Fm}$ | 0.031 | 0.025 | 0.033 |
| MRT (y) | 7.58±3.5 | 66.9±23 | 143±57 |
| $CV_{MRT\,(y)}$ | 0.46 | 0.34 | 0.40 |
| SOC (%) | 37.93±11.4 | 11.68±7.26 | 6.40±4.66 |
| $CV_{SOC\,(\%)}$ | 0.30 | 0.62 | 0.73 |



**Table 3.** Intra-plot variability for three spatial variability end-members, expressed in $\Delta^{14}C$ and standard deviation across regions and horizons converted to mean residence time (MRT) and coefficient of variation (CV). Variability is the highest for resp. the Podzol and Gleysol, lowest for the Cambisol.

|  | Podzol 0-5 cm (n =7) | Gleysol 0-5 cm ( n = 8) | Cambisol 0-5 cm ( n = 8) |
|---|---|---|---|
| $\Delta^{14}C$ (‰) | −1.48±43.26 | 126±35.2 | 129±18.5 |
| Fm | 1.00±0.047 | 1.13±0.038 | 1.35±0.020 |
| $CV_{Fm}$ | 0.047 | 0.033 | 0.018 |
| MRT (y) | - | 73.0±27 | 2.64±2.1 |
| $CV_{MRT}$ | - | 0.37 | 0.78 |



**Table 4.** Variability in $^{14}$C signature expressed in Fm and CV for lateral (plot) and horizontal (depth) variation of five different soils.

| | Podzol | | | Gleysol | | | Calcisol | | Stagnosol | | | Cambisol | | |
|---|---|---|---|---|---|---|---|---|---|---|---|---|---|---|
| | 0-5 | 5-10 | 10-20 | 0-5 | 5-10 | 10-20 | 0-5 | 10-20 | 0-5 | 5-10 | 10-20 | 0-5 | 5-10 | 10-20 |
| **Fm lat** (n =4) | 0.980 ±0.012 | 0.860 ±0.017 | 0.816 ±0.035 | 1.12± 0.019 | 1.08± 0.018 | 0.909 ±0.17 | 1.07±0.034 | 1.02±0.008 | 1.14±0.005 | 1.08±0.013 | 1.03±0.014 | 1.15± 0.016 | 1.03± 0.016 | 1.00± 0.007 |
| **CV Fm lat** (n =4) | 0.014 | 0.015 | 0.007 | 0.017 | 0.017 | 0.190 | 0.032 | 0.008 | 0.004 | 0.012 | 0.013 | 0.014 | 0.015 | 0.007 |
| | **Topsoil depth** | **Subsoil depth** | | **Topsoil depth** | **Subsoil depth** | | **Topsoil depth** | **Subsoil depth** | **Topsoil depth** | **Subsoil depth** | | **Topsoil depth** | **Subsoil depth** | |
| **CV Fm hor** | 0.988±0.032 | 0.0887±0.018[3] | | 1.08±0.10 | 1.03±0.14[4] | | 1.04±0.065 | 0.978±0.10[5] | 1.10±0.049 | 1.02±0.17[6] | | 1.06±0.048 | 1.05±0.059[7] | |
| **CV Fm hor** | 0.032 | 0.019 | | 0.097 | 0.13 | | 0.063 | 0.11 | 0.044 | 0.17 | | 0.045 | 0.056 | |

[3] 0-65 cm
[4] 0-40 cm
[5] 0-60 cm
[6] 0-70 cm
[7] 0-50 cm



**Table 5.** Spearman correlation of $\Delta^{14}C$ versus climatic variables. Significant correlations are found between $\Delta^{14}C$ against MAP and SOC at 10-20 cm. For p-values $p < 0.05$, $p < 0.005$ and $p < 0.0005$, values correlations are indicated with *, ** and ***, $p > 0.05$ is indicated with superscript ns.

| | MAT | MAP | NPP | Elevation | Slope | SOC |
|---|---|---|---|---|---|---|
| Organic layer (n =10) | $-0.12^{ns}$ | $-0.38^{ns}$ | $-0.47^{ns}$ | $0.02^{ns}$ | $-0.07^{ns}$ | $-0.37^{ns}$ |
| 0-5 cm depth (n =10) | $0.29^{ns}$ | $0.10^{ns}$ | $0.08^{ns}$ | $0.04^{ns}$ | $0.65*$ | - |
| 5-10 cm depth (n =11) | $-0.55^{ns}$ | $-0.21^{ns}$ | $0.02^{ns}$ | $0.27^{ns}$ | $-0.2^{ns}$ | $-0.29^{ns}$ |
| 10-20 cm depth (n = 15) | $0.12^{ns}$ | $-0.78***$ | $0.12^{ns}$ | $-0.12^{ns}$ | $0.31^{ns}$ | $-0.59*$ |



**Table 6.** Results of the mixed effect model testing $\Delta^{14}C$ for the effects of environmental variables. Site and core were always taken as random effects, whilst log(SOC), Elevation, MAT, MAP NPP, pH and log(Clay) are taken separately as fixed effects. The table shows F-values and significant effects are indicated are indicated with *, ** and *** for p-values $p < 0.05$, $p < 0.005$ and $p < 0.0005$. p-values $> 0.05$ are indicated with superscript ns.


| | Log(SOC) | Elevation | MAT | MAP | NPP | pH_H20 | Log(Clay) |
|---|---|---|---|---|---|---|---|
| 0-5 cm (n=12 sites, n = 39 samples) | 5.9* | $3.8^{ns}$ | 4.9* | $0.11^{ns}$ | $0.97^{ns}$ | $0.0022^{ns}$ | $0.29^{ns}$ |
| 5-10 cm (n=4 sites, n = 15 samples) | $0.23^{ns}$ | $2.3^{ns}$ | $1.1^{ns}$ | $0.27^{ns}$ | $1.5^{ns}$ | $0.34^{ns}$ | $1.3^{ns}$ |
| 10-20 cm (n=5 sites, n = 20 samples) | 14** | $0.41^{ns}$ | $0.12^{ns}$ | $0.64^{ns}$ | $0.53^{ns}$ | $0.43^{ns}$ | $0.32^{ns}$ |