# Peer review of "Variability in 14C contents of soil organic matter at the plot and regional scale across climatic and geologic gradients"

_Biogeosciences, 2015_

## Referee Comment (RC1) · Anonymous Referee #1 · 24 Jan 2016

Review of Biogeo Disc / van der Voort Radiocarbon is a valuable tool in attempts to understand the formation and turnover of soil organic matter, but the difficulty and expense of making measurements have meant that the available data are relatively few. Therefore the authors' idea to assess the variability and representativeness of 14C measurements is welcome. Broadly, they conclude that there is not a great deal of spatial variation, so that depth variations in 14C are fairly consistent from site to site, and results already available have therefore been reasonably representative. I do not think this is at all a trivial conclusion – it is an important finding. My main criticism of the work is that the variable chosen to represent soil carbon was SOC concentration (%), which is not directly relevant to SOM turnover. Better would be SOC pool (g/m2),

which is the natural relative of turnover rate. Therefore I suggest that either the authors justify the use of concentrations, or they reanalyse their results using C pools. More minor comments: Line 21 Here the results for "topsoils" are claimed to have been reported. As far as I understand it by topsoil they mean the top of the mineral soil beneath the O (LF) layer. I question whether this really is topsoil in the sense of containing organic matter undergoing turnover and being intimately connected to ecosystem processes – in other words I reckon that the LF material is functionally important and should be counted as soil. If the authors do not agree, then some discussion would be welcome. It would be of interest to know for example how much C (g/m2) is in the LF layer, and how much in the 0-5 cm at the top of the mineral soil. Moreover, the numbers quoted in the Abstract ($\Delta$14C 159, sd 36.4) appear actually to refer to the LF layer (Table 2)! Line 26 This last line of the Abstract is weak, if there are "important consequences" you should say what they are. Materials and methods The dates of much of the soil sampling are stated to be "in the course of the 1990s" which could mean that some samples were collected 10 years apart. Other samples were collected in 2014. It therefore is not strictly correct to compare 14C values, since they are not constant with time in soil in situ – indeed that is why 14C is a useful variable, and why the data of sampling is an important qualifier of every 14C measurement. Maybe the analysis here would not be much affected by the assumption that the 14C values refer to the same point in time, but the issue should be acknowledged and the assumption justified – perhaps the MRT values are sufficiently long that a few years' difference in sampling date is of no consequence? Equation (2) This doesn't look right to me – the leading $\Delta$14C shouldn't be there. Also, is it really necessary to apply the equation only to samples with a value of R>1, which is what seems to be stated in lines 124-5? And after reading further, I realise that I do not understand the difference between R and Fm. Line 173 I don't see why the expressions "worst case" and "best case" are used here – the facts are the facts, we should not judge them. Also, I do not fully understand what is learnt by showing that the variability of 14C correlates (or does not correlate) with variables like slope, MAP etc. This is not

considered in the Discussion, yet the results for variation with clay and MAP appear as conclusions. I could not see any information about clay contents (e.g. why not in Table 1?). Line 234 The word "marked" here is used rather carelessly. The values of MAP and MAT admittedly vary, but within fairly small ranges in a global context. And since the soil types and geologies also vary it can hardly be claimed that variations in the site attributes have been sufficiently covered – it might be for example that a trend in MAT counters one in MAP, or in soil type or in geology, or indeed in vegetation type (as far as I can see no information on tree species is provided, certainly not in Table 1) or NPP. Although the results are certainly of considerable interest, the fact that definite trends cannot be found does not mean that there are no trends Line 288 Is it really necessary to incorporate "factors that drive small-scale variability" into larger-scale models of SOM turnover? Is it not possible that ecosystem complexity and the costs of analysis mean that the more complex models implied here are unachievable?

Please also note the supplement to this comment:
http://www.biogeosciences-discuss.net/bg-2015-649/bg-2015-649-RC1-supplement.pdf

---

## Referee Comment (RC2) · S. Trumbore (Referee) · 25 Jan 2016

This study reports a large set of data collected on soil C and radiocarbon profiles from a range of sites in Switzerland. Soil studies are often conducted according to 'state factor' analysis; i.e. climate- or chrono- sequences. However, the important of state factors should hopefully also fall out of statistical analyses that include sample covering a wide range of state factors. This study has two goals: (1) to analyze spatial variability of C and 14C within single "sites", and (2) to document spatial variability among different soils that vary in factors like temperature, precipitation, NPP, etc. for forested soils across Switzerland. There is an exploration of how microtopography may be responsible for spatial variability within forest plots; this showed very interesting

differences in variability for different soil types (e.g. Podzols versus Cambisols).

Overall, the study provides a valuable data set, especially very few data are available that deal with the spatial heterogeneity of SOC and 14C and how that might impact comparisons made at broader spatial scales.

As with other studies documenting variability across sites (e.g. Schrumpf et al. 2013, Herold et al. (2014) and Mathieu et al. (2015), the variations in the vertical are always larger than variations laterally for 14C (and C). Although the soils studied differ in many respects (e.g. parent material geology, climate, etc), all are apparently quite young soils (developed on moraines or outwash fans). This is pointed out in the paper (lines 329-330), but perhaps could be highlighted a bit more than it is as an explanation for similarity among soil profiles.

The authors should add more information to Table 1, including total soil depth - are these also all shallow soils, or do the soils continue deeper than the depth-specific sampling? Although the authors investigated the predictive capability of a number of factors, such as clay content, pH, etc., the reader never knows the range of these values (they are not given in Table 1, please give at least a profile average here for the factors used in the multi-regression). Maybe the lack of difference (except for the Podzols) arises from the overall similarity in these factors of many of the soils studied? The differences in C content would seem to indicate not, but the reader is not able to judge.

Also, although all of these are forested sites, is there any evidence that they were previously unforested (e.g. Ap plow layers)?

A second issue that affects variability is something like the presence or absence of earthworms (for example, these tend to be found in Cambisols but not in Podzols, and they also affect the thickness and age of C in the litter layer. The 'biota' state factor includes in-soil fauna, it could account for some of the differences in variability among the different soil types. Normally such things are noted in profile descriptions, and are

semiquantitative; nonetheless they may be important.

Similar findings regarding similarity of vertical profiles of 14C in different soils were obtained by Mathieu et al 2015, which came out around the time this was submitted; while 14C characteristics are similar at the surface, deeper soils reflect the influence of soil order (something that can be related to geology and vegetation/climate regime and time together). However, that study used global soils, and mixed in with soil order is soil age (there are not young oxisols, or old inceptisols). A more comparable study to this one would be Schrumpf et al. 2013, which is cited here but it would be interesting to compare their estimates of spatial variability with yours (as a function of depth).

The use of %C as the metric for C content is problematic, especially in litter layers, which can have highly variable bulk density. Is there information to report carbon density gC cm-2 for each of the depth intervals?

Some more detailed comments:

Line 119. Were samples stored in glass jars or paper bags?

Lines 150-155. If the 14C signature of bulk C was above the contemporary atmosphere 14C, there will be two solutions (two values of k) that can reproduce that value with a single pool model. Which one did you choose, and what reasoning did you use to decide? This needs to be described in the paper.

Line 172. When you say variables such as clay content, pH, etc were taken as "fixed effects", does that mean you used some profile-averaged value in statistical comparisons? I found this description confusing, can you make it clearer? Also, please give the values for pH, clay, etc in Table 1. If available, cation exchange capacity might also be a useful variable.

Line 271. Schrumpf et al. (2013) found a relationship between the slope of the radiocarbon-depth relationship and dithionite extractable Fe; Herold et al. (2014) also found that Fe(d) was a good predictor of C content. This indicates that a common

stabilization mechanism may be operating across their soils, which could also be an explanation for the similarly of depth profiles. Is there any similar measure for these soils (even cation exchange capacity, which is more frequently measured than Fe(d))?

Line 293-4. The link of 14C to MAP as reflecting waterlogging is a bit speculative at the larger spatial scales, though you do have possible evidence from the intra-site variability in soils that have evidence of redox variability (e.g. Figure 6). But at larger spatial scales, would not clay content be expected to be related to drainage (e.g. does this relationship trace to Gleysols and Stagnosols?)

The next lines, about relief, are also a bit speculative. How was "relief' reported in Table 1 determined? At the microtopographic scale, or the macrotopographic scale? While I agree it may indicate something about erosion in general, it may also be correlated with other factors like parent material, temperature, etc. You need a separate measure (e.g. 137Cs) to say something like this definitively.

Line 303. Typo, should be "noted"

I did not understand lines 304-305: " but when assuming a steady state system, it is reasonable to assume that the speed of incorporation of carbon and hence turnover is directly related to carbon stocks." Do you mean the larger the C stock the faster the turnover should be (e.g. as it is with soil depth, most C and fastest C at the surface?) or do you mean the more 'standard' sense, of largest stocks having overall slowest turnover (e.g. integrating low C concentration over the large volume of deep soil means it has the largest stock, which is associated with slowest turnover). This is a place where it is important to give C stocks, not just concentrations.

Line 334 " the relative independence on climatic parameters may persist in deeper soils" However, you did have a relationship with MAP – which could indicate some kind of effect of redox-related stabilization (see above). Overall, stabilization mechanisms appear to operate on similar timescales, independent of the amount of C being stabilized?

[Figure]

The discussion of microtopography is a little frustrating for the reader to follow, as there is never really a good definition of what the authors mean by it. We can visualize 'hummocks' and 'hollows', but can their spatial dimensions be better quantified? Were they really traceable to tree-throw? Or perhaps (in young soils) to variations in the underlying till structure (e.g. the presence of a large underlying boulder)?

Lines 374-378. How were the semivariograms constructed? Did you try to use a specific depth (e.g. 0-5 cm) or integrated depth profiles (e.g. kgC m-2, or C-weighted mean 14C)? Would it make a difference? (perhaps soil depths also vary, but this was not captured in your sampling scheme..)

Lines 386-7. Soils subjected to fluctuating redox conditions might be expected to overall cycle C faster (if the major stabilization mechanisms have to do with Fe-oxides). Also, sampling across mottles (reduced and oxidized Fe) can mix C of quite different ages (see Fimmen et al. 2008 )

Line 390. "Overall, the geochemical characteristics..." You have mentioned only one indicator, the C/N ratio. This is a good indicator of decomposition in organic layers, but I am not convinced it is so good deeper in the mineral soil (though you are mixing different stabilization mechanisms together, low-density and mineral-associated material). It would be nice to have some factors that more directly relate to stabilization mechanisms themselves (e.g. cation exchange capacity, or surface area; see Lawrence et al. 2015).

Lines 408-9. "the speed of C incorporation may be relatively insensitive to changing climate conditions" However all soils had bomb C – so the speed of C incorporation is relatively fast overall; it is just that it is similarly fast. Also, you do have evidence of sensitivity in the factors that create microtopography (erosion/redox variation) both of which can change with climate conditions.

Figure 3. Error bars for the vertical axis (%SOC) are not visible – are they small or just not shown?

Figure 4B. It is apparent that the Nitrex site used to study microtopography ( is not sampled at constant depth intervals; in other words, 14C samples are integrating different depth intervals. Thus, especially for the deepest horizon, it is difficult to see that the resulting trends are due to microtopography rather than sampling (lowest 14C has the largest integrated depth interval). Or am I missing the intent of this figure?

References

Fimmen, R. L., Richter, D. D., Vasudevan, D., Williams, M. A., & West, L. T. (2008). Rhizogenic Fe-C redox cycling: A hypothetical biogeochemical mechanism that drives crustal weathering in upland soils. Biogeochemistry, 87(2), 127-141. 10.1007/s10533-007-9172-5

N Herold, I Schöning, B Michalzik, S Trumbore, M Schrumpf (2014) Controls on soil carbon storage and turnover in German landscapes. Biogeochemistry 119 (1-3), 435-451

Mathieu, J. A., Hatté, C., Balesdent, J. and Parent, É. (2015), Deep soil carbon dynamics are driven more by soil type than by climate: a worldwide meta-analysis of radiocarbon profiles. Glob Change Biol, 21: 4278–4292. doi:10.1111/gcb.13012

M Schrumpf, K Kaiser, G Guggenberger, T Persson, I Kögel-Knabner, et al (2013) Storage and stability of organic carbon in soils as related to depth, occlusion within aggregates, and attachment to minerals. Biogeosciences 10, 1675-1691

---

## Referee Comment (RC3) · Anonymous Referee #2 · 9 Feb 2016

The paper explores variations in 14C contents of soil organic matter within and between forest sites in Switzerland. It is an interesting, experimentally well done, and clearly described piece of work and will be useful in the design of future studies using 14C to trace turnover times of soil organic matter. There is little that I can add to what has already been mentioned in the two other reviews.

Perhaps it is a matter of taste. In many instances in the text I would replace the adjective "high" with "large" or "great" (e.g. line 47: "large carbon stock" instead of "high carbon stock").

---

## Author Comment (AC1) · 12 Apr 2016

Dear Referee,

First of all thank you very much for taking time to review this paper and for the valuable comments and suggestions that you have made. Below you will find th unformatted text and replies, but we have attached a pdf with formatting.

Review of Biogeo Disc / van der Voort Radiocarbon is a valuable tool in attempts to understand the formation and turnover of soil organic matter, but the difficulty and expense of making measurements have meant that the available data are relatively few. Therefore the authors' idea to assess the variability and representativeness of

14C measurements is welcome. Broadly, they conclude that there is not a great deal of spatial variation, so that depth variations in 14C are fairly consistent from site to site, and results already available have therefore been reasonably representative. I do not think this is at all a trivial conclusion – it is an important finding. My main criticism of the work is that the variable chosen to represent soil carbon was SOC concentration (%), which is not directly relevant to SOM turnover. Better would be SOC pool (g/m2), which is the natural relative of turnover rate. Therefore I suggest that either the authors justify the use of concentrations, or they reanalyse their results using C pools.

–> Thank you for the positive feedback –> We did not include the SOC pools originally because our bulk density (BD) estimates are not considered to be very precise (as they are based on single profiles taken proximally to the plot, see Walthert et al. 2002, 2003). However, based on this suggestion, we have incorporated available BD data and now values for SOC stocks have been incorporated in our statistical analysis (Table 5, 6). However, as the C stocks (gC/cm3) are the multiplication of C content (gC/g soil) with BD (g soil/cm3) and soil interval length, no new additional correlations emerge in our statistical analysis.

More minor comments: Line 21 Here the results for "topsoils" are claimed to have been reported. As far as I understand it by topsoil they mean the top of the mineral soil beneath the O (LF) layer. I question whether this really is topsoil in the sense of containing organic matter undergoing turnover and being intimately connected to ecosystem processes – in other words I reckon that the LF material is functionally important and should be counted as soil. If the authors do not agree, then some discussion would be welcome.

–> Indeed, by topsoil we mean the top of the mineral soil beneath the O(LF) layer. It is correct that the LF material is functionally important, and it was also incorporated in this paper, although it receives less focus than the mineral soil. It was measured in a number of locations (Fig. 3.) to get a better idea of variability. We calculated turnover for this layer (Table 2). We also did statistical analysis on it (Variability analysis in Table

2, Spearman correlation in Table 5).

It would be of interest to know for example how much C (g/m2) is in the LF layer, and how much in the 0-5 cm at the top of the mineral soil.

–> For the 0-5 cm layer we have bulk-density estimated from single profiles proximal to the plot where the samples were taken. For the LF layer we found data that we were previously unaware of collected on the same plot, and this enabled us to determine the carbon stocks (WSL LWF). These data are now included in Supplemental Table 1.

Moreover, the numbers quoted in the Abstract ($\triangle$14C 159, sd 36.4) appear actually to refer to the LF layer (Table 2)!

–> this was corrected - thank you.

Line 26 This last line of the Abstract is weak, if there are "important consequences" you should say what they are.

–> This formulation indeed broad, as our conclusions that apply to models are two-fold. On the one hand, large small-scale heterogeneity could be incorporated in plot-scale or catchment-scale models (CENTURY, DAYCENT etc), whilst on the other hand, for large spatial scale (Earth system models) we have a relative homogeneity. Because of this, we would prefer to keep this statement generic and provide more specific details later on in the manuscript.

Materials and methods The dates of much of the soil sampling are stated to be "in the course of the 1990s" which could mean that some samples were collected 10 years apart. Other samples were collected in 2014. It therefore is not strictly correct to compare 14C values, since they are not constant with time in soil in situ – indeed that is why 14C is a useful variable, and why the data of sampling is an important qualifier of every 14C measurement. Maybe the analysis here would not be much affected by the assumption that the 14C values refer to the same point in time, but the issue should be acknowledged and the assumption justified – perhaps the MRT values are sufficiently

long that a few years' difference in sampling date is of no consequence?

–> Thank you for this suggestion; this is absolutely correct. The measurements on these sites that have been done in 2014 are included in another paper (Van der Voort et al., in prep) for which the different atmospheric signal will be taken into account. This paper only concerns data only of the period 1994-1998. Because this was unclear, we have removed the reference of this sampling campaign.

Equation (2) This doesn't look right to me – the leading $\triangle$14C shouldn't be there. Also, is it really necessary to apply the equation only to samples with a value of R>1, which is what seems to be stated in lines 124-5? And after reading further, I realise that I do not understand the difference between R and Fm.

–> Thank you for that remark, indeed there was a typo, which has been corrected.

Line 173 I don't see why the expressions "worst case" and "best case" are used here – the facts are the facts, we should not judge them.

–> Thank your for suggestion, we have adjusted the wording to incorporate this suggestion:

Line 199: "ranges from 50 ‰ (relative highest degree of variability scenario, Podzol) to 20 ‰ (relative lowest degree of variability, Cambisol) (Table 3).

Also, I do not fully understand what is learnt by showing that the variability of 14C correlates (or does not correlate) with variables like slope, MAP etc. This is not considered in the Discussion, yet the results for variation with clay and MAP appear as conclusions.

–> This is a good point, and required clarification. We have clarified this is the results section: line 176: "The Spearman coefficient identified few significant correlations between ïĄĎ14C in samples and climatic variables". Further details: (1) This paper looks at the variability on different scales, such as the plot and regional scale (Tables 2, 3 and 4). We do compare it to slope in the text, as we considered this as a causal factor in

the degree of variability. (2) We then look at the correlation between ∆14C and Slope, MAP etc. (Tables 5 & 6), because we would like to know if SOM dynamics depend on these environmental variables

I could not see any information about clay contents (e.g. why not in Table 1?).

–> Thank you for this suggestion, this has been added in Supplement Table 1.

Line 234 The word "marked" here is used rather carelessly. The values of MAP and MAT admittedly vary, but within fairly small ranges in a global context. And since the soil types and geologies also vary it can hardly be claimed that variations in the site attributes have been sufficiently covered – it might be for example that a trend in MAT counters one in MAP, or in soil type or in geology, or indeed in vegetation type (as far as I can see no information on tree species is provided, certainly not in Table 1) or NPP. Although the results are certainly of considerable interest, the fact that definite trends cannot be found does not mean that there are no trends

–> This is a valid point. We tried to eliminate the site-induced variability by inserting an nmle linear mixed-effect model and taking the Site/Core variable as the random variable. However, it is correct that this dataset has its limitations w.r.t. MAT and MAP range, and we explicitly tackle this by saying: Line 369 "While the present observations remain limited in geographic scope, the relative homogeneity of ∆14C signatures observed in surface and deep soils across climatic and geologic gradients implies that the rate of C incorporation may be similar and hence relatively insensitive to changing climate conditions".

Line 288 Is it really necessary to incorporate "factors that drive small-scale variability" into larger-scale models of SOM turnover? Is it not possible that ecosystem complexity and the costs of analysis mean that the more complex models implied here are unachievable?

–> For global models, this point is absolutely valid. There are however plenty of

plot-scale models (i.e. CENTURY, DAYCENT, YASSO), for which these factors could be taken into account. We have adjusted the wording to make this point clearer: Line 380 "The latter is essential for the use of radiocarbon to assess carbon turnover and associated processes in forest soils, especially for plot-scale modelling."

Please also note the supplement to this comment:
http://www.biogeosciences-discuss.net/bg-2015-649/bg-2015-649-AC1-supplement.pdf

―――――――――――――――――――

---

## Author Comment (AC2) · 12 Apr 2016

Dear Prof. Trumbore,

Many thanks for the valuable input on this paper, it is much appreciated. We have attached a supplement pdf with formatting which may be easier to read. In the following we provide responses to your comments and suggestions:

"As with other studies documenting variability across sites (e.g. Schrumpf et al. 2013, Herold et al. (2014) and Mathieu et al. (2015), the variations in the vertical are always larger than variations laterally for 14C (and C). Although the soils studied differ in many respects (e.g. parent material geology, climate, etc), all are apparently quite young

soils (developed on moraines or outwash fans). This is pointed out in the paper (lines 329-330), but perhaps could be highlighted a bit more than it is as an explanation for similarity among soil profiles. "

–> thank you for this suggestion, we have incorporated this point in: line 282 "Soil formation for the soils studied here initiated after the last glacial retreat and can hence be assumed to have started to form simultaneously, which may explain the similarity in their 14C distribution with depth."

–> we have also incorporated the Mathieu paper (which indeed came out as this was submitted). Line 256 "Mathieu et al., (2015) also found that the carbon dynamics in deeper soils are not controlled by climate but rather by pedologic traits, whereas topsoil carbon dynamics were found to be related to climate and cultivation."

The authors should add more information to Table 1, including total soil depth - are these also all shallow soils, or do the soils continue deeper than the depth-specific sampling? Although the authors investigated the predictive capability of a number of factors, such as clay content, pH, etc., the reader never knows the range of these values (they are not given in Table 1, please give at least a profile average here for the factors used in the multi-regression). Maybe the lack of difference (except for the Podzols) arises from the overall similarity in these factors of many of the soils studied? The differences in C content would seem to indicate not, but the reader is not able to judge.

–> thank you for this suggestion. We have added this information in Supplement Table 1 as there was insufficient space in the original table. (1) to Supplement table 1 has been added. a. Clay, sand, silt, pH content of the 0-5 cm layer average of a single profile. The 0-5 cm layer was chosen as it was the most heavily studied. b. Range of total soil depth as determined during the 1990's sampling of the WSL LWF campaign c. Bulk density in a single profile for the 0-5 cm layer as determined in the field. Description of method can be found in method section. d. Carbon stocks for 0-5 cm layer

calculated by multiplication of density (gsoil/cm3) × carbon concentration (gC/gsoil) × layer length (cm) = (gC/m2). This data was not included initially because the bulk density measurements were derived from single profiles. But as you requested we now added it. The bulk density method is now described in the methods section:

Line 89 "These samples were taken using steel cylinders of 1000 cm3 volume (for layers with a thickness of at least 10 cm) or 458 cm3 volume (for thin layers with less than 10 cm thickness). Volumetric samples were dried at 105 °C for 48 hours minimum until the resulting mass remained constant. The density of the fine earth was determined based on oven-dried volumetric soil samples and sieving the samples in a water bath to quantify the weight of stones >2 mm. The volume of stones was calculated by assuming a density of 2.65 kg/m3 for stones (Walthert et al., 2002)."

Also, although all of these are forested sites, is there any evidence that they were previously unforested (e.g. Ap plow layers)?

–> thank you for this suggestion, these sites have been forests for at least several hundreds of years (Gosheva et al., in prep, personal communication) as shown in older Swiss maps. Furthermore, the sites in the WSL LWF campaign are specifically selected to be mature forests (LWF).

A second issue that affects variability is something like the presence or absence of earthworms (for example, these tend to be found in Cambisols but not in Podzols, and they also affect the thickness and age of C in the litter layer. The 'biota' state factor includes in-soil fauna, it could account for some of the differences in variability among the different soil types. Normally such things are noted in profile descriptions, and are semiquantitative; nonetheless they may be important.

–> thank you for this suggestion. We found information on soil biota in some of these sites that we were previously unaware of (Ernst et al., 2008), and have incorporated this suggestion:

Line 271: "Furthermore, the presence of soil fauna (earthworms) at some sites (Bettlachstock, Schaenis, Lausanne, Alptal, Visp and Novaggio) may also complicate the response of carbon cycling to climate due to physical reworking and transport (Ernst et al., 2008)."

Line 325: "Ernst et al (2008) described the presence of earthworms in the Gleysol and Cambisol, but not in the Podzol. Because of constraints on the dataset size, no conclusive quantitative relationship can be established, but we hypothesise that the ubiquitous presence of in-soil fauna and associated transport activities would contribute to an overall increase in homogeneity rather than heterogeneity."

Similar findings regarding similarity of vertical profiles of 14C in different soils were obtained by Mathieu et al 2015, which came out around the time this was submitted; while 14C characteristics are similar at the surface, deeper soils reflect the influence of soil order (something that can be related to geology and vegetation/climate regime and time together). However, that study used global soils, and mixed in with soil order is soil age (there are not young oxisols, or old inceptisols).

–> Thank you for this suggestion, the Mathieu paper indeed came out as this manuscript was submitted, we have incorporated it as stated above. Line 256 "Mathieu et al., (2015) also found that the carbon dynamics in deeper soils are not controlled by climate but rather by pedologic traits, whereas topsoil carbon dynamics were found to be related to climate and cultivation."

A more comparable study to this one would be Schrumpf et al. 2013, which is cited here but it would be interesting to compare their estimates of spatial variability with yours (as a function of depth).

–> Schrumpf et al. (2013) HF and oLF values fall within the same $\Delta$14C range the profiles measured in this paper (Fig.8). Because the Schrumpf et al. (2013) paper only refers to values of the fractions and in this current paper only bulk is concerned, we chose not to include a direct comparison. We will be sure to include this information in

planned future papers that include fraction-specific radiocarbon data.

The use of %C as the metric for C content is problematic, especially in litter layers, which can have highly variable bulk density. Is there information to report carbon density gC cm-2 for each of the depth intervals?

–> (1) Although this is not part of the normal WSL LWF database, we were able to acquire information of the approximate litter layer bulk density, and we have now included it in Supplement Table 1. (2) Additionally, we have added estimated carbon stocks (N.B., the bulk density is determined based on a single profile proximal to the plot where samples described here were collected), and have included them in the linear mixed effects models. (Supplement Table 1, extended Table 5, 6)

Line 119. Were samples stored in glass jars or paper bags?

–> Samples in the WSL Pedotheque are stored in plastic containers. This has been incorporated: lines 64-65: "The LWF sites are all located in mature forests and samples were collected in the during the 1990s and have been stored in plastic containers (Innes, 1995)"

Lines 150-155. If the 14C signature of bulk C was above the contemporary atmosphere 14C, there will be two solutions (two values of k) that can reproduce that value with a single pool model. Which one did you choose, and what reasoning did you use to decide? This needs to be described in the paper.

–> This is a very good point. In the cases where two options were possible we chose the option which corresponded with the turnover estimates of the layers above and below, as we assume deeper soil layers to always have slower turnover than shallower soil layers: Line 135 "For the 0-5 cm topsoil layer two MRT were frequently possible, in which case it was assumed the true MRT value of the deeper layer is the one that exceeds the MRT value of the accompanying litter layer, as carbon turnover rates decrease with increasing soil depth."

[Figure]

Line 172. When you say variables such as clay content, pH, etc were taken as "fixed effects", does that mean you used some profile-averaged value in statistical comparisons? I found this description confusing, can you make it clearer? Also, please give the values for pH, clay, etc in Table 1. If available, cation exchange capacity might also be a useful variable. –> This indeed required clarification. First, we used the clay and pH values measured for the layer depth interval identical to that which was measured. Hence, it is not a profile average but sample-depth specific value. We have clarified this in the text: line 156 "The compositional parameters (e.g. clay, pH) are depth interval-specific."

–> w.r.t. cation Exchange capacity: We found more ancillary data that we were previousy unaware of, from which we calculated the CEC (after Blume et al (2002), Lehrbuch Bodemkunde chapter 5) and have now included that in the linear mixed effect models analysis.

Line 271. Schrumpf et al. (2013) found a relationship between the slope of the radiocarbon-depth relationship and dithionite extractable Fe; Herold et al. (2014) also found that Fe(d) was a good predictor of C content. This indicates that a common stabilization mechanism may be operating across their soils, which could also be an explanation for the similarly of depth profiles. Is there any similar measure for these soils (even cation exchange capacity, which is more frequently measured than Fe(d))?

–> We also found ancillary data for Fe and other metals (Fe, Al) extracted by $HNO_3$. within the WSL LWF database for this, and have included it in the linear mixed effect model. However, only in the 0-5 cm layer the linear mixed-effect model indicates a significant positive relation between $\Delta 14C$ and Fe content, for all other depths the correlation is not significant.

Line 293-4. The link of 14C to MAP as reflecting waterlogging is a bit speculative at the larger spatial scales, though you do have possible evidence from the intra-site variability in soils that have evidence of redox variability (e.g. Figure 6). But at larger

spatial scales, would not clay content be expected to be related to drainage (e.g. does this relationship trace to Gleysols and Stagnosols?)

–> Since we only have two sites to compare in this context we feel that we cannot test this, but we adjusted the wording to more accurately reflect the nature of the statement. Line 258 "The strong negative Spearman correlation of $\Delta$14C and MAP at 10-20 cm depth implies a slower turnover that could potentially be caused by increased water-logging or anoxic conditions induced by higher precipitation."

The next lines, about relief, are also a bit speculative. How was "relief' reported in Table 1 determined? At the microtopographic scale, or the macrotopographic scale? While I agree it may indicate something about erosion in general, it may also be correlated with other factors like parent material, temperature, etc. You need a separate measure (e.g. 137Cs) to say something like this definitively.

–> The slope in table 1 was determined on a scale of the larger WSL LWF sites, i.e., several tens of meters in both directions. –> The relief of two sites (Lausanne, lowest variability and Beatenberg, highest variability) have been monitored and the curvature of the larger area has been calculated using ArcGIS. This information was not available for Alptal. From this we can quantitatively observe that the degree of variability varies significantly. The surface in the Lausanne plot hardly has any curvature whilst Beatenberg has strong irregular microtopographic oscillations. This will be added to the supplemental documentation. Line 303. Typo, should be "noted" incorporated

I did not understand lines 304-305: "but when assuming a steady state system, it is reasonable to assume that the speed of incorporation of carbon and hence turnover is directly related to carbon stocks." Do you mean the larger the C stock the faster the turnover should be (e.g. as it is with soil depth, most C and fastest C at the surface?) or do you mean the more 'standard' sense, of largest stocks having overall slowest turnover (e.g. integrating low C concentration over the large volume of deep soil means it has the largest stock, which is associated with slowest turnover). This is a place

where it is important to give C stocks, not just concentrations.

–> This issue was addressed by adding the C stocks. We have clarified the formulation. We meant the "standard" sense, i.e. that larger stocks are associated with slower turnover. Furthermore, the Spearman correlation between MRT and C stocks in the Litter layer gave a strongly significant positive relation (0.77**) indicating that larger stocks are associated with a higher MRT thus slower turnover. This will be incorporated into the results section. Line 275 "in a steady state system it is reasonable to assume slower turnover is coupled to larger carbon stocks".

Line 334 "the relative independence on climatic parameters may persist in deeper soils" However, you did have a relationship with MAP – which could indicate some kind of effect of redox-related stabilization (see above). Overall, stabilization mecha- nisms appear to operate on similar timescales, independent of the amount of C being stabi- lized?

–> This is a valid point. We have adjusted the wording to fit more appropriately: line 309: "the relative independence on temperature and primary production may persist in deeper soils" This paper does not provide a detailed discussion on stabilisation mech- anisms, as it focuses on $\Delta 14C$ and less, for instance, on organo-mineral interactions, but in future work are seeking to also examine this.

The discussion of microtopography is a little frustrating for the reader to follow, as there is never really a good definition of what the authors mean by it. We can visualize 'hummocks' and 'hollows', but can their spatial dimensions be better quantified? Were they really traceable to tree-throw? Or perhaps (in young soils) to variations in the underlying till structure (e.g. the presence of a large underlying boulder)?

–> Thank you for the suggestion: (1) The description for the Gleysol is as quantitative as possible, with descriptions of mound/depression height and width of the mounds and depression. Unfortunately, no ancillary data is presently available. We agree that could be better to have radar images of the surface, but acquiring that is beyond the

scope of our present project. (2) For the Cambisol and Podzol we have curvature plots, which have been added to the appendix. (3) Tree-throw has been observed visually in the field. (4) We do not think variations in till structure play a role as we took numerous cores in 2014 from the same sites and did not see any significant structural variation. Again, however, we can only provide a qualitative indication.

Lines 374-378. How were the semivariograms constructed? Did you try to use a specific depth (e.g. 0-5 cm) or integrated depth profiles (e.g. kgC m-2, or C-weighted mean 14C)? Would it make a difference? (perhaps soil depths also vary, but this was not captured in your sampling scheme..)

–> The Semivariograms were only for the 0-5 cm interval because available spatial variability data was most abundant for that depth. For deeper samples, we do not have sufficient data, but this would certainly be interesting to look into in the future.

Lines 386-7. Soils subjected to fluctuating redox conditions might be expected to overall cycle C faster (if the major stabilization mechanisms have to do with Fe-oxides). Also, sampling across mottles (reduced and oxidized Fe) can mix C of quite different ages (see Fimmen et al. 2008)

–> Thank you for the helpful suggestion. Indeed, Fimmen et al (2008) found a positive correlation between changing redox conditions with an increase in C breakdown (especially when Fe is high such as in this site). In our case the intermediate system (mottled) has the oldest signal, which assuming the results of Fimmen (2008) are true, would indicate that this system would be under more stable redox conditions, as opposed to the stronger depression. From the topography and groundwater flow, we can only suppose that the deeper the soil the more permanently it would be submerged, i.e. we would expect the deepest soil to have the most stable redox conditions. However, we do not know enough about the groundwater flow to make any conclusive statements. We do not think it is likely to be a local mottled/non-mottled effect, as the sample is the average over a depth interval and several cores. Given the Fimmen et

al. (2008) results are inconsistent with our results, and that our evidence is too inconclusive to go against their conclusions, we have left this discussion out of this present contribution.

Line 390. "Overall, the geochemical characteristics..." You have mentioned only one indicator, the C/N ratio. This is a good indicator of decomposition in organic layers, but I am not convinced it is so good deeper in the mineral soil (though you are mixing different stabilization mechanisms together, low-density and mineral-associated material). It would be nice to have some factors that more directly relate to stabilization mechanisms themselves (e.g. cation exchange capacity, or surface area; see Lawrence et al. 2015).

–> CEC has been included in the Supplemental Table 1. Unfortunately surface area information is not available, for future work we try to acquire this information.

Lines 408-9. "the speed of C incorporation may be relatively insensitive to changing climate conditions" However all soils had bomb C – so the speed of C incorporation is relatively fast overall; it is just that it is similarly fast. –> Good point, we have adjusted the wording: line 368 "the speed of C incorporation may be similar and hence relatively insensitive to changing climate conditions."

Also, you do have evidence of sensitivity in the factors that create microtopography (erosion/redox variation) both of which can change with climate conditions. Indeed, potentially, with changing climate, extreme weather events like storms and droughts could be more commonplace, which in turn could induce stronger microtopography by forming rills etc. This remains very speculative however, and from the available data we cannot really say what effect that would have on the long-term cycling of soil carbon, except to point out that increased extreme events may increase erosion. As this remains speculative, we have not incorporated this discussion into the paper. Obtaining data pertinent to this question this would be a valuable line of future research.

Figure 3. Error bars for the vertical axis (%SOC) are not visible – are they small or just

not shown?

–> they are smaller than the point.

Figure 4B. It is apparent that the Nitrex site used to study microtopography ( is not sampled at constant depth intervals; in other words, 14C samples are integrating different depth intervals. Thus, especially for the deepest horizon, it is difficult to see that the resulting trends are due to microtopography rather than sampling (lowest 14C has the largest integrated depth interval). Or am I missing the intent of this figure?

–> Indeed, the four types in the NITREX plot serve to assess variability. The horizon-specific sampling method has an advantage because it also allows assessment of the effect of microtopography on horizon development and morphology (Fig. 6), but it is indeed disadvantageous because impedes a direct quantitative comparison with the other study sites. While we can only compare the signals qualitatively, it gives us a better idea on variability on a catchment-wide scale. The specific separate microtopographic effects are shown in Fig. 6.

Please also note the supplement to this comment:
http://www.biogeosciences-discuss.net/bg-2015-649/bg-2015-649-AC2-supplement.pdf

---

## Author Comment (AC3) · 12 Apr 2016

Dear Referee,

Thank you for the reply and positive feedback, it is very much appreciated. We have adjusted adjectives where appropriate.